# BCAT1 controls metabolic reprogramming in activated human macrophages and is associated with inflammatory diseases

Adonia E. Papathanassiu[1], Jeong-Hun Ko[2], Martha Imprialou[2], Marta Bagnati[2], Prashant K. Srivastava[3], Hong A. Vu[1], Danilo Cucchi[4,5], Stephen P. McAdoo[6], Elitsa A. Ananieva[7], Claudio Mauro[4] & Jacques Behmoaras[2]

Branched-chain aminotransferases (BCAT) are enzymes that initiate the catabolism of branched-chain amino acids (BCAA), such as leucine, thereby providing macromolecule precursors; however, the function of BCATs in macrophages is unknown. Here we show that BCAT1 is the predominant BCAT isoform in human primary macrophages. We identify ERG240 as a leucine analogue that blocks BCAT1 activity. Selective inhibition of BCAT1 activity results in decreased oxygen consumption and glycolysis. This decrease is associated with reduced IRG1 levels and itaconate synthesis, suggesting involvement of BCAA catabolism through the IRG1/itaconate axis within the tricarboxylic acid cycle in activated macrophages. ERG240 suppresses production of IRG1 and itaconate in mice and contributes to a less proinflammatory transcriptome signature. Oral administration of ERG240 reduces the severity of collagen-induced arthritis in mice and crescentic glomerulonephritis in rats, in part by decreasing macrophage infiltration. These results establish a regulatory role for BCAT1 in macrophage function with therapeutic implications for inflammatory conditions.

[1] Ergon Pharmaceuticals, LLC, P.O. Box 1001, Silver Spring, Maryland 20910, USA. [2] Centre for Complement and Inflammation Research, Imperial College London, London W12 0NN, UK. [3] Division of Brain Sciences, Imperial College Faculty of Medicine, London W12 0NN, UK. [4] William Harvey Research Institute, Barts and the London School of Medicine and Dentistry, Queen Mary University of London, London EC1M 6BQ, UK. [5] Institute Pasteur, Fondazione Cenci Bolognetti, Rome 00161, Italy. [6] Renal and Vascular Inflammation Section, Department of Medicine, Imperial College London, London W12 0NN, UK. [7] Biochemistry and Nutrition, Des Moines University, Des Moines, Iowa 50312, USA. Correspondence and requests for materials should be addressed to A.E.P. (email: adoniap@ergon-pharmaceuticals.com) or to C.M. (email: c.mauro@qmul.ac.uk) or to J.B. (email: jacques.behmoaras@imperial.ac.uk).

Branched-chain aminotransferase (BCAT) is the enzyme responsible for the reversible transamination of leucine, isoleucine and valine, the amino acids collectively known as branched-chain amino acids (BCAA). Transamination of BCAAs is the first step in their catabolism and results in the formation of branched-chain α-keto acids, which are decarboxylated to form derivatives of coenzyme A (CoA)[1,2]. In the case of leucine, transamination leads to the formation of α-ketoisocaproate, which is further metabolized to form the ketone body acetoacetate, and acetyl-CoA, which is subsequently oxidized in the tricarboxylic acid (TCA) cycle. This reaction equally results in the production of glutamate, which is another crucial metabolite that feeds into the TCA cycle at the level of α-ketoglutarate[3]. Hence, BCAT enzymes are multi-level regulators of the TCA cycle and oxidative phosphorylation, possibly contributing to metabolic reprogramming in eukaryotic cells.

BCAT exists in two isoforms, mitochondrial BCAT2 and cytosolic BCAT1. Although *Bcat2* is expressed ubiquitously in most tissues and especially in skeletal muscle, tissues of the digestive system and the kidney, *Bcat1* expression is reported to be limited to embryonic tissues, adult brain, ovary, placenta and neurons of the peripheral nervous system[4]. Despite the large body of work on BCAA metabolism and, in particular, Bcat1 activity in glutamate neurotransmitter metabolism in the brain[4] and in tumour growth[5], data on the role of BCAT1 and its potential metabolic effects on macrophages and inflammatory disease in general do not exist.

Macrophages are innate immune cells with a phenotype tightly linked to their metabolism[6–9]. They have remarkable plasticity in function, with both extreme and intermediate activation phenotypes[10,11] under specific transcriptional control[12]. Toll-like receptor agonist lipopolysaccharide (LPS) stimulation results in a rapid and robust transcriptional response that involves genes that regulate metabolic reprogramming[13]. Macrophages activated by LPS undergo a metabolic shift from oxidative phosphorylation to glycolysis known as the Warburg effect[9,14], with some important metabolic effectors regulating the balance between the two metabolic states[15]. The increased consumption of glucose is known to be associated with a proinflammatory macrophage phenotype[16]. Similar metabolic reprogramming has been detected in mouse dendritic cells, with an early response to LPS corresponding to an increase in glycolytic rate within minutes of exposure[17]. Macrophages activated by LPS also show accumulation of Krebs cycle intermediates, such as succinate, regulating Hif-1α-mediated IL-1β production[14]. A comprehensive integrative analysis of the transcriptome and metabolome in activated macrophages has established the importance of citrate-itaconic acid axis through Irg1 (also known as cis-aconitate decarboxylase, Acod1)[18], one of the most significantly up-regulated transcripts in LPS-stimulated macrophages, which encodes an enzyme that catalyses the aconitate-to-itaconate reaction[18,19]. These studies led to the concept of the broken (or fragmented) Krebs cycle in LPS-activated macrophages (M(LPS))[18,20], in which TCA cycle intermediate metabolites function as metabolic checkpoints for the activation of LPS response genes, such as *Hif1a*, *Il1b*, *Irg1*. Furthermore, these data show that reprogramming macrophage metabolism is not only needed for energy requirements, but also for the transcriptional regulation of the innate immune response.

By using high-throughput RNA measurements in human monocyte-derived macrophages, we show that BCAT1 is the most abundantly expressed BCAT isoform. Using a novel selective inhibitor of BCAT1 (ERG240, a leucine analogue), we show that inhibition of BCAT1 leads to the reduction of oxygen consumption and glycolysis together with a decrease in IRG1 and itaconate, but not HIF-1α and IL-1β messenger RNA (mRNA)

and protein levels; suggesting induction of BCAT1-mediated anaplerosis through the leucine–citrate–itaconic acid axis in early macrophage activation with LPS. To confirm the effect of ERG240 *in vivo*, we show that peritoneal macrophages from mice injected with LPS and ERG240 have reduced levels of Irg1/itaconate together with a diminished proinflammatory transcriptome when compared with mice injected with LPS alone. RNAi of *BCAT1* in human M (LPS) phenocopies the effect of pharmacological blockade with reduced BCAT1 protein levels resulting in reduction of oxygen consumption and glycolysis together with decreased IRG1 and itaconate levels. We then investigate the anti-inflammatory effects of ERG240 in two distinct murine models of autoimmune disease (rheumatoid arthritis and crescentic glomerulonephritis) characterized by macrophage activity and infiltration. We show that ERG240 treatment results in reduced inflammation associated with decreased macrophage infiltration in target organs (joints and kidney) in both models.

## Results

**BCAT1 regulates metabolic reprogramming in macrophages.** In a previous genome-wide expression quantitative trait loci analysis using inbred rat models of glomerulonephritis, we identified *Bcat1* as the hub of a co-expression network in macrophages, suggesting its potential regulatory metabolic role in inflammatory disease[21]. To gain more insights into the role of BCAT enzymes in macrophages, we set out to investigate these in human monocyte-derived macrophages (hMDMs) by subjecting these cells to RNA-sequencing (Fig. 1a). This showed that mRNA levels of *BCAT1* are markedly increased when compared to *BCAT2*, suggesting that the cytoplasmic isoform is the major source of BCAA catabolism in these cells. To examine the role of BCAT1 in macrophage metabolism and inflammatory diseases, we identified a novel BCAT1 inhibitor, ERG240, using molecular modelling approaches based on the X-ray crystal structures of BCAT1 (ref. 22). ERG240 is a water-soluble structural analogue of leucine (Fig. 1b). Its ability to suppress the BCAT1 aminotransferase activity was measured in a continuous fluorometric assay (ref. 23 and see Supplementary Methods). We found that ERG240 inhibited recombinant BCAT1 with an $IC_{50}$ of 0.1–1 nM, while no inhibition was observed on BCAT2 (Fig. 1c). The effect of BCAT1 inhibition on hMDMs was studied in control (basal, M(basal)) or LPS-stimulated cells by measuring mRNA and protein levels of three major TCA-regulated early macrophage activation markers: HIF-1α, IL-1β (ref. 14) and IRG1 (ACOD1)[18]. As expected, LPS treatment caused the up-regulation of HIF-1α, IL-1β and IRG1 at mRNA and protein levels (Fig. 1d,e). Strikingly, treatment with ERG240 (3 h, 20 mM) resulted in a dramatic reduction in *IRG1* mRNA levels but not *HIF1A* and *IL1B* in LPS-stimulated cells (Fig. 1d). These results were also confirmed at the IRG1 protein levels in LPS and ERG240-treated cells (Fig. 1e). Consequently, the ERG240-mediated reduction in IRG1 levels resulted in decreased itaconate levels measured by GC/MS (Fig. 1f). ERG240 had comparable inhibitory effect on *IRG1* expression in TNF-stimulated human macrophages (Supplementary Fig. 1A) and long-term exposure (24 h) to LPS and ERG240 resulted in reduced IL-1β levels (Supplementary Fig. 1B), suggesting a temporal effect of ERG240 on IL-1β. Furthermore, short-term exposure of ERG240 significantly reduced *TNF* and *NOS2* but not *IL6* and *PTGS2* expression levels in human macrophages (Supplementary Fig. 1F). ERG240 did not affect cell viability at 3 and 8 h LPS stimulation in neither control nor LPS-treated human macrophages (Supplementary Fig. 1E). Importantly, the effect of BCAT1 inhibition on IRG1 expression and itaconate

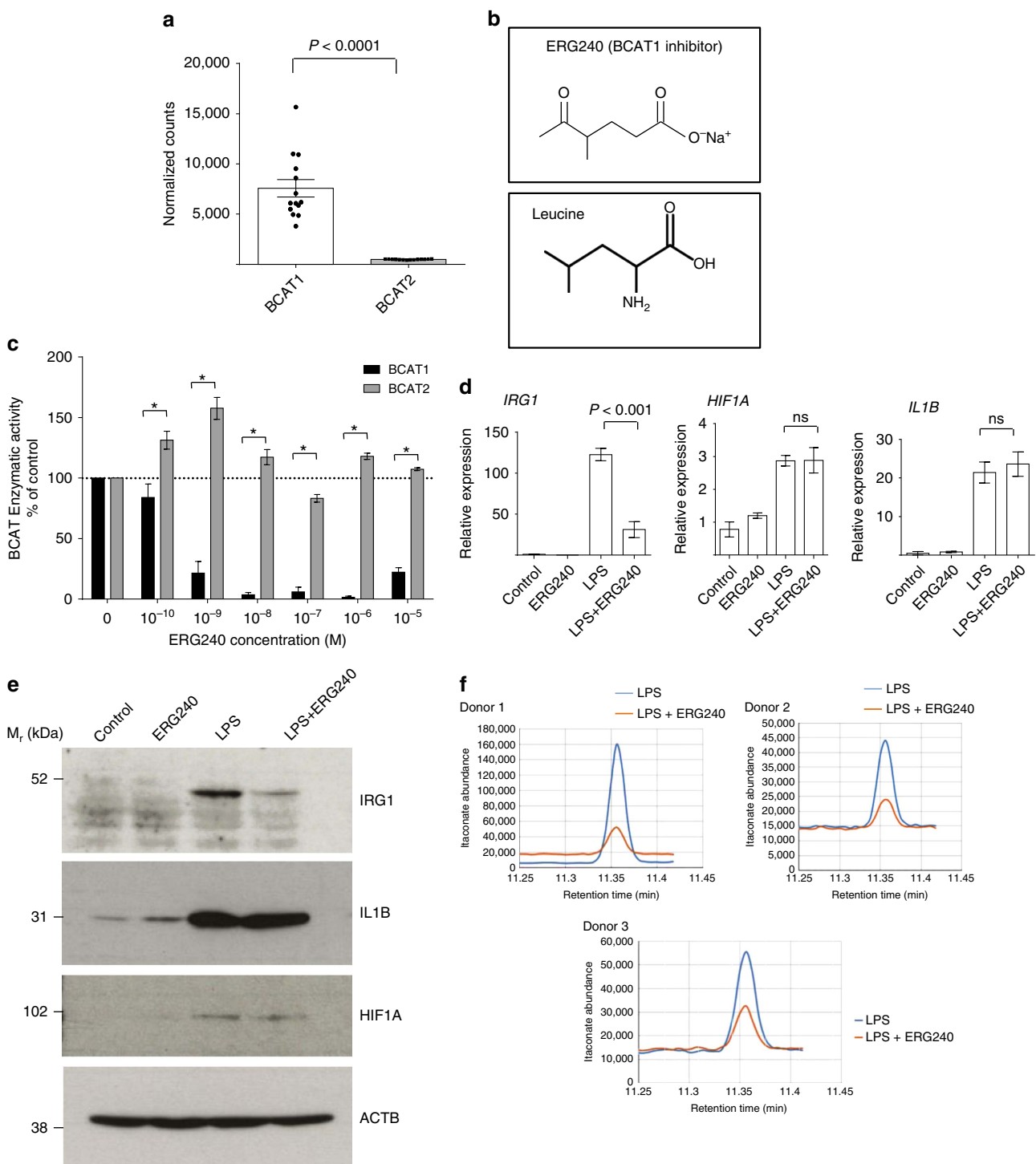

**Figure 1 | BCAT1 expression and inhibition in human macrophages. (a)** RNA-sequencing in human monocyte-derived macrophages (hMDMs) shows mRNA copies of BCAT enzymes in normalized counts. $n = 14$ healthy donor hMDMs were used for RNA-seq. **(b)** Chemical structure of ERG240 and leucine. **(c)** Enzymatic activity of recombinant human BCAT1 and BCAT2 in presence of ERG240. **(d)** Relative expression of *IRG1*, *HIF1A* and *IL1B* measured by qRT-PCR in control, ERG240-treated (20 mM, 3 h), LPS treated (100 ng ml$^{-1}$, 3 h), and LPS + ERG240 treated (LPS, 100 ng ml$^{-1}$; ERG240, 20 mM for 3 h) hMDMs. ns, non-significant. All expression values are normalized to those obtained for *HPRT* gene expression. $n = 3$ healthy donor hMDMs were used in each group. **(e)** Western blot analysis of IRG1, IL-1β, HIF-1α and beta-actin (ACTB) in control (untreated), ERG240-treated (20 mM, 3 h), LPS treated (100 ng ml$^{-1}$, 3 h) and LPS + ERG240 treated (LPS, 100 ng ml$^{-1}$; ERG240, 20 mM for 3 h) hMDMs. The experiment is representative of three independent experiments using $n = 3$ healthy donor hMDMs each. **(f)** GC/MS results showing itaconic acid production in hMDMs in presence or absence of ERG240 (20 mM). LPS treatment was for 8 h (100 ng ml$^{-1}$). GC/MS plot for each donor is shown separately where itaconate abundance denotes arbitrary units. Error bars are s.e.m. Significance was tested using two-tailed Student's *t*-test or one-way ANOVA. * $P < 10^{-3}$ following two-way ANOVA.

levels was ascertained in macrophages stimulated with LPS in a BCAA-deprived media (Supplementary Fig. 1C and D), confirming the requirement for an enzymatically active BCAT1 for its inhibition of IRG1-mediated itaconate production.

Next, we measured the effect of BCAT1 inhibition on oxygen consumption and glycolysis in hMDMs. The mitochondrial respiration and glycolytic metabolism were assessed by measurement of oxygen consumption rate (OCR) and extracellular acidification rates (ECAR), respectively. The bioenergetic profiles were determined by sequential use of oligomycin (mitochondrial ATP synthase inhibitor), FCCP (protonophore and uncoupler of ATP synthesis from mitochondrial respiration), and a combination of rotenone, antimycin A and 2-deoxyglucose (2-DG) that allows the inhibition of mitochondrial respiration and glycolysis. LPS treatment alone resulted in increased OCR (basal respiration, estimated ATP production and maximal respiration) and ECAR (glycolysis and glycolytic capacity) in hMDMs (Supplementary Fig. 2). Treatment of LPS-stimulated hMDMs with ERG240 resulted in decreased OCR and ECAR (Fig. 2a,b). This translated into a significantly reduced basal respiration, estimated ATP production, maximal respiration, glycolysis and glycolytic capacity (Fig. 2a,b). In keeping with the decreasing effect of ERG240 on oxygen consumption and glycolysis, siRNA-mediated inhibition of *BCAT1* mRNA levels resulted in a significant decrease in OCR and ECAR levels (Fig. 2c,d). We also measured citrate by GC/MS in hMDMs as the breakpoint TCA metabolite in M(LPS)[18] and found a relative increase in citrate levels following LPS stimulation, which was further significantly increased in the presence of ERG240 (Supplementary Fig. 3A), suggesting that the effect of BCAT1 inhibition on the broken TCA cycle in human macrophages lies predominantly downstream citrate at the IRG1 site with a relatively minor effect through inhibition of BCAA transamination and reduction in the amount of acetyl-coA entering the cycle (Supplementary Fig. 3B). To confirm the pharmacological inhibition with ERG240, we used RNAi on human BCAT1 to measure IRG1 protein levels and itaconate (Fig. 3a). The experiment showed consistent knock down of BCAT1 associated with reduced IRG1 levels in three independent donors (Fig. 3a). In keeping with the pharmacological blockade, BCAT1 knockdown resulted in reduced IRG1 and itaconate levels in hMDMs (Fig. 3b).

**ERG240 suppresses *in vivo* production of IRG1/itaconate.** To confirm the inhibitory effect of ERG240 on IRG1-mediated itaconate production in M(LPS) *in vivo*, we injected C57BL/6 mice with either LPS (1.5 mg kg$^{-1}$) or LPS in conjunction with ERG240 (500 mg kg$^{-1}$), and harvested peritoneal macrophages 24 h later. These results showed a reduction in *Irg1* mRNA and protein levels associated with a significant decrease in itaconate (Fig. 4a–c). Itaconate was equally decreased when ERG240 was administered 3 h after the first LPS injection (Fig. 4c).

To gain more insights into the effect of ERG240 on macrophage polarization, we performed RNA-sequencing (RNA-seq) in vehicle, LPS and LPS + ERG240 treated peritoneal macrophages at the time of collection (24 h). When the top 100 (97 annotated, 28 up-regulated and 72 down-regulated) differentially expressed transcripts ($10^{-310} <$ FDR $< 10^{-73}$) between LPS and LPS + ERG240 treated groups were analysed by STRING for high confidence protein-protein interactions (interaction score = 0.7), this led to the identification of an interferon (IFN)-inducible GTPase pathway (PFAM protein domains FDR = $6.97 \times 10^{-6}$; Fig. 4d). Interestingly, all of the 25 genes belonging to the IFN-inducible GTPase signature are down-regulated together with M1-like markers (*Nos2*, *Il6*), while M2-like and extracellular matrix transcripts are up-regulated in the peritoneal macrophages obtained from ERG240-treated mice

(Fig. 4e-g). *Mrc1*, *Klf4* and *Cd36* were previously described as alternative (M2-like) macrophage activation markers[24–26], and Fragments Per Kilobase of transcript per Million fragments mapped (FPKM) values for these transcripts show significant up-regulation in ERG240-treated macrophages (Fig. 4g). *Ifit* and *Gbp* gene families' expression values are also illustrated in Fig. 4g. Furthermore, the extracellular matrix enrichment is striking (Cellular component GO: extracellular space, FDR = $1.04 \times 10^{-8}$) among the 28 most significantly up-regulated genes. FPKM values. FPKM values for *Fn1*, *Mmp9*, *Gsn* and *Spp1* are shown in Fig. 4g. Taken together, these results show that long-term ERG240 exposure in LPS-injected mice is associated with a reduced proinflammatory and increased anti-inflammatory and repair transcriptome signature in peritoneal macrophages.

**ERG240 reduces the severity of immune-mediated inflammation.** First, a dose-escalating toxicity study was performed and indicated that no major adverse effects were associated with the oral or intraperitoneal administration of ERG240 when the drug was given at doses up to 2 g kg$^{-1}$. A subsequent pharmacokinetics study revealed that ERG240 showed excellent bioavailability with a relatively short half-life in plasma suggesting either rapid absorption or clearance of the drug (Tables 1 and 2).

The reduced proinflammatory transcriptome signature following treatment with ERG240 in the peritoneal macrophages of LPS-injected mice led us hypothesize that BCAT1 blockade could be beneficial in broader inflammatory conditions. To test the potential anti-inflammatory effects of ERG240 in a macrophage-dependent model of immune complex-driven tissue inflammation, we assessed the effect of ERG240 administration in the well-established nephrotoxic nephritis (NTN) model utilizing the uniquely susceptible Wistar-Kyoto (WKY) rat strain[27,28]. The macrophage-dependent kidney inflammation observed in this model resembles the immune complex mediated glomerular damage in systemic lupus erythematosus referred as lupus nephritis[29]. Oral administration of ERG240 during 10 days following NTN induction resulted in significantly reduced glomerular crescent formation, proteinuria, and serum creatinine (Fig. 5a), demonstrating the ability of ERG240 to reduce the severity of inflammation in a highly reproducible and severe model of antibody-mediated kidney inflammation. Treatment with ERG240 did not interfere with the deposition of the nephrotoxic serum to the glomerular basement membrane. Quantitative immunofluorescence for collagen type I alpha I (Cola1) in nephritic kidneys (day 28) showed a significant down-regulation in the ERG240-treated animals (Fig. 5b), which was in accordance with the reduced Sirius red staining and *Col1a1* mRNA levels in the renal cortex (Fig. 5b). These results showed that treatment with ERG240 resulted in reduced interstitial fibrosis in the NTN model of crescentic glomerulonephritis.

In addition to NTN, macrophages play a central role in the pathogenesis of rheumatoid arthritis (RA) where they infiltrate synovial membranes[30,31], contributing to chronic inflammation. Bone erosion in RA has also been linked to monocytes/macrophages since these cells are known to differentiate into osteoclasts in the presence of M-CSF and RANKL[32]. Hence, the effect of Bcat1 inhibition on collagen-induced arthritis (CIA) in mice was examined prophylactically, by administering 720 mg kg$^{-1}$ ERG240 at the time of the booster injection 21 days after the initial immunization, and therapeutically, by administering 1000 mg kg$^{-1}$ ERG240 after arthritis was established. The results show that oral administration of ERG240 alleviates the severity of CIA (Fig. 5c). Analysis of the sera obtained before immunization and at the end of the

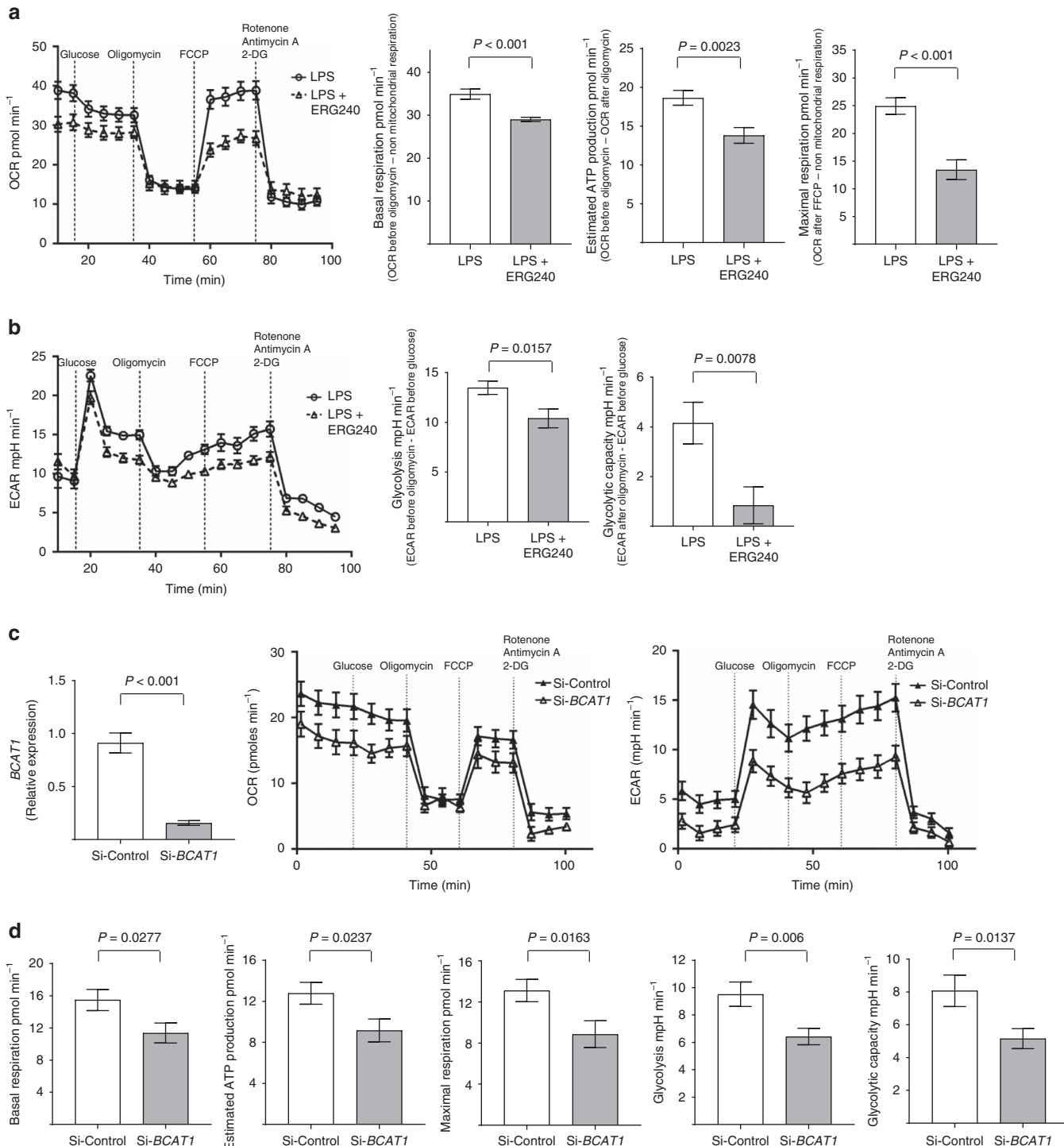

**Figure 2 | BCAT1 inhibition reduces oxygen consumption and glycolysis in human macrophages.** (**a**) Real-time extracellular OCR measurements in hMDMs (left panel) that were previously treated with either LPS (100 ng ml$^{-1}$) or LPS + ERG240 (20 mM) for 3 h. OCR values were recorded following sequential treatments with glucose (5 mM), oligomycin A (0.5 μM), carbonyl cyanide 4-(trifluoromethoxy) phenyl- hydrazine (FCCP, 1 μM) and a combination of 2-deoxy-D-glucose (2-DG, 50 mM), antimycin A (1 μM) and rotenone (1 μM). Basal respiration, estimated ATP production, and maximal respiration are shown in the right panel. $n = 3$ healthy donor hMDMs were used in 12 technical replicates. The results are representative of three independent experiments. (**b**) Real-time ECAR (left panel), glycolysis and glycolytic capacity (right panel). $n = 3$ healthy donor hMDMs were used in 12 technical replicates. The results are representative of three independent experiments. (**c**) hMDMs were subjected to *BCAT1* siRNA (si-*BCAT1*) or non-targeting siRNA (Si-Control) treatment, followed by LPS stimulation (3 h, 100 ng ml$^{-1}$) before measuring *BCAT1* expression levels (left panel), OCR (middle panel) and ECAR (right panel). (**d**) Basal respiration, estimated ATP production, maximal respiration, glycolysis and glycolytic capacity in si-Control and si-*BCAT1* transfected, LPS-treated hMDMs. $n = 4$ hMDMs were used in at least five technical replicates. Error bars are s.e.m. Significance was tested using two-tailed Student's *t*-test.

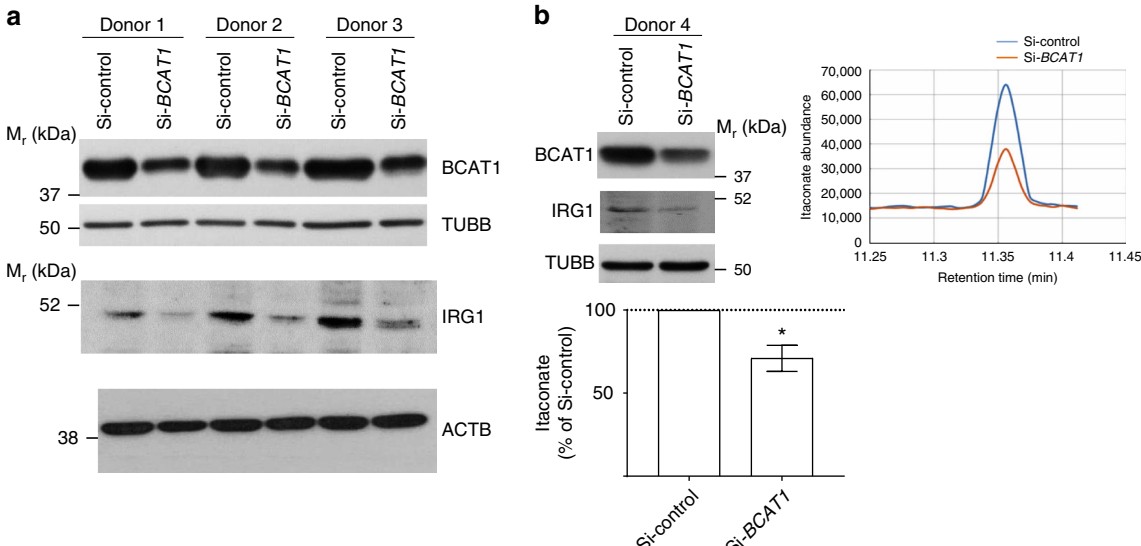

**Figure 3 | BCAT1 silencing leads to reduced IRG1 and itaconate levels in activated human macrophages. (a)** Western Blot following BCAT1 (Si-*BCAT1*) and non-targeting (Si-Control) siRNA treatment in hMDMs stimulated with LPS (100 ng ml$^{-1}$, 8 h). BCAT1, IRG1, TUBB and ACTB blots are shown on three independent donor hMDMs. **(b)** BCAT1 and IRG1 Western Blot on Si-*BCAT1* and Si-Control hMDMs from donor four and corresponding itaconate levels measured by GC/MS are shown in the right panel. Itaconate levels measured by GC/MS were further confirmed in si-*BCAT1* and si-Control hMDMs from independent donors (n = 4, lower panel). Error bars are s.e.m. Significance was tested using one sample *t*-test (two-tailed). *P < 0.05 by one-sample-*t*-test.

experiment validated the presence of antibodies against type II collagen and confirmed that ERG240 did not interfere with antibody production (Supplementary Figure 4).

Following completion of CIA, excised paws were analysed for presence of inflammation, pannus formation, cartilage damage and bone resorption by immunohistochemistry (Fig. 5c). Despite its moderate effect on oedema formation, ERG240 markedly reduced the progression of the disease by decreasing inflammation by 53%, cartilage degradation by 74% and pannus formation and bone erosion by 86% (Fig. 5c). We found a significant decrease in joint oedema with ERG240 administration only in the fore paws of the animals in the therapeutic study (Supplementary Fig. 5). We further measured serum Tnf and Tnfsf11 (RANKL) levels in CIA mice and found diminished levels of these circulating markers upon ERG240 treatment (Supplementary Fig. 6).

**ERG240 treatment reduces macrophage infiltration *in vivo*.** We first evaluated the potential effect of ERG240 on macrophage migration and showed that ERG240 suppresses BMDM migration in a dose-dependent manner and with an IC$_{50}$ value of ~5–10 mM (Fig. 6a). Inhibition of migration was not incidental to a cell death event as ERG240 did not affect cell numbers after treatment of BMDM for 24 h (Fig. 6a). To further investigate the mechanisms of action of ERG240 *in vivo*, excised paws were tested for the presence of F4/80 by immunohistochemistry and we found that there was a significant reduction in F4/80$^+$ macrophages in the joints following ERG240 treatment (Fig. 6b). Similarly to what we observed in CIA, macrophage infiltration was significantly reduced in nephrotoxic nephritis model as the glomerular infiltration of CD68 (ED1)-positive cells were significantly decreased in ERG240-treated animals (Fig. 6c).

**Discussion**
A growing number of studies suggest that cellular metabolism, immune response and inflammation are intimately linked processes under common, evolutionary conserved regulatory pathways[33]. The majority of the reports supporting the above

concept focus on glucose, glutamine and lipid metabolism or document an association between deregulated metabolism and immune pathology[7,34,35]. BCAA catabolism provides generally a mechanism for regulating the levels of leucine, isoleucine and valine, whose excess has been previously associated with neural dysfunction[36,37]. Up-regulation of Bcat1 under pathological conditions may serve to regulate the local levels of BCAAs and especially leucine, which is known to activate the mammalian target of rapamycin complex 1 (mTORC1). In CD4+ T cells, Bcat1 regulates mTORC1 signalling and glycolytic metabolism, highlighting the important role of leucine metabolism[38]. BCAA transamination serves also as the source of amino groups for the synthesis of non-essential amino acids such as glutamine and alanine[39]. BCAT1 is considered as a secondary target of gabapentin, a common antiepileptic and antinociceptive drug that is known to bind with high affinity to $a_2\delta-1$ and $a_2\delta-2$ subunits of voltage-activated calcium channels[40]. However, gabapentin has been previously shown to display no apparent anti-inflammatory effects[41]. The role of BCAA catabolism in macrophage function is poorly described. In a recent report, Meiser *et al.* observed an increased uptake of leucine in RAW264.7 murine macrophage cell line stimulated with LPS both in normoxia and hypoxia[42], suggesting that LPS stimulation in macrophages results in an increase of BCAA as an alternative carbon source to glucose and glutamine[42].

Here, we describe BCAT1 as the predominantly expressed BCAT isoform in human macrophages. We found that pharmacological blockade of BCAT1 resulted in decreased oxygen consumption and glycolysis in LPS-treated macrophages. We focused on early LPS activation (3 h) because major transcriptome changes involving primary response genes occur 2-4 h following LPS stimulation in macrophages[13,43] and monocytes[44] with some key Toll-like receptor4-dependent markers such as *Hif1a* (ref. 45), *Il1b* (refs 43,46) and *Irg1* (refs 19,47) detected as a rapid transcriptional activation response. All these early response markers were found to be tightly regulated by TCA in M(LPS)[14,18], in particular *Irg1* and its enzymatic product, itaconate, which accumulates following 8–10 h LPS stimulation in primary mouse macrophages[48] and

RAW264.7 cells[19]. Itaconate is a bactericidal metabolite that could have distinct roles depending on the pathological context[49]. Besides its antimicrobial effects, itaconate has also been proposed as an inhibitor of substrate level phosphorylation through its effect on succinate-CoA ligase in RAW-264.7 cells[50]. Furthermore, itaconic acid has been shown to inhibit SDH[48,51,52] and a recent study linked IRG1-itaconic acid production to modulation of M2-like macrophage polarization in the revascularization of ischaemic muscle[53]. Investigating pathways upstream and downstream IRG1/itaconate are the focus of recent studies[48,52,54]

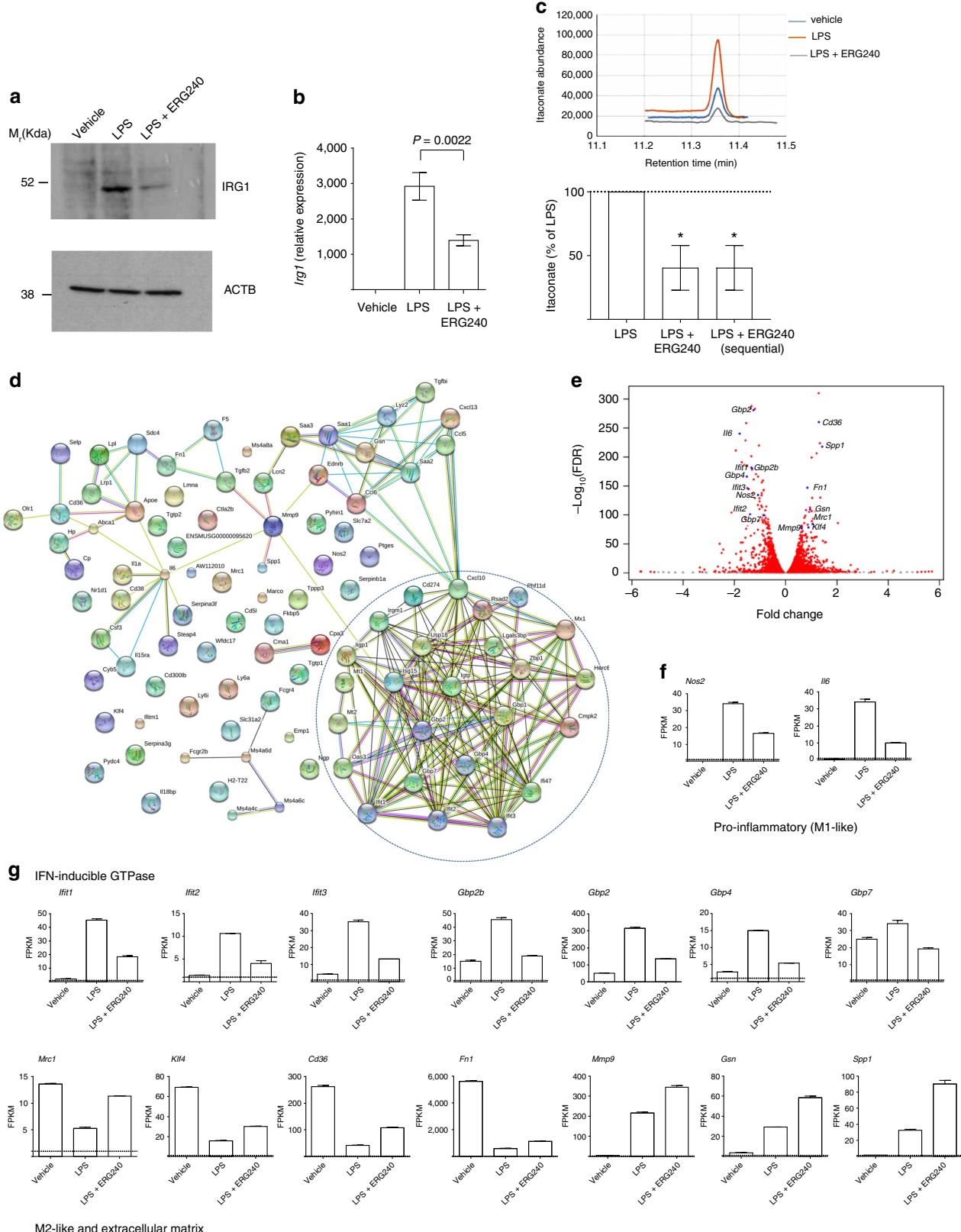

and will facilitate understanding the gene-metabolite links in activated macrophages. This will be crucial in determining the mechanisms by which BCAT1 regulate itaconate synthesis.

Metabolism in M(LPS) macrophages is characterized by high levels of glycolysis, a fragmented TCA and relatively high levels of some of its intermediates (succinate, citrate and itaconate). By contrast, metabolism in M2 (or M(IL-4)) corresponds to high levels of fatty acid oxidation (FAO) and oxidative phosphorylation even though the exact role of FAO in macrophage activation states require further clarification[55]. Our findings indicating (i) reduced ECAR levels as a result of BCAT1 inactivation in human macrophages and (ii) anti-inflammatory effects of ERG240, reversing the M1-like transcriptome in mouse peritoneal macrophages, argue in favour of the involvement of BCAA in M1/proinflammatory macrophage function. Although the metabolic response of murine macrophages and dendritic cells to LPS stimulation has been thoroughly explored[15,17], the bioenergetic profile of human macrophages is not well described and recent reports suggest that culturing conditions (M-CSF versus GM-CSF) play an important role in their metabolic reprogramming[56]. Here, we report that early LPS stimulation of human macrophages resulted in an increase of ECAR, which is indicative of increased glycolytic rate and is consistent with previous results obtained in murine bone marrow derived and peritoneal macrophages[15].

The recently discovered discontinuity in the TCA cycle at the Isocitrate Dehydrogenase 1 (IDH1) level, resulting in a flux re-direction towards citrate/aconitate has important implications in terms of proinflammatory macrophage activation[18]. This metabolic axis enables large production of fatty acids and itaconate and our results suggest that leucine catabolism through BCAT1 is a regulator of IRG1-dependent macrophage activation. It is likely that BCAT1 regulates the production of fatty acids and, in accordance with this, its expression levels in human adipose tissue were found to correlate with weight regain, making BCAT1 expression the best marker for this phenotype[57]. Likewise, a down-regulation in BCAT1 expression was found to correlate with a dietary weight loss program[58].

Here, we demonstrate that Bcat1 is a druggable target for controlling inflammation in two distinct chronic inflammatory diseases characterized by antibody-mediated end-organ damage with macrophage infiltration. The rat model of crescentic glomerulonephritis in the WKY strain is dependent on macrophage infiltration and activation in response to glomerular deposition of immunoglobulin[21,59,60]. Primary macrophages from the WKY strain show a genetically determined Hif-1α-mediated glycolytic transcriptome signature during their differentiation[61], suggesting a genetically determined metabolic state that could explain their unique susceptibility to glomerular inflammation. In RA, macrophages have been described as the primary pathogenic drivers of the disease with fibroblasts, T and B lymphocytes playing secondary roles[62]. The protective effect of Bcat1 inhibition in two LPS-independent inflammatory models suggest that Bcat1 may have its regulatory role on macrophages activated with different proinflammatory stimuli and this is consistent with the in vitro observation that ERG240 inhibits Irg1 mRNA levels in TNF-activated human MDMs. Nevertheless, the role of macrophage-derived BCAT1 in inflammatory models can only be firmly established by using conditional targeted gene deletion of Bcat1 in monocytes/macrophages.

The druggability of Bcat1 in autoimmune disease was shown with the utilization of ERG240, a novel Bcat1 inhibitor. ERG240 had an overall favourable bioavailability profile with no observable toxicity and, although administered at high concentrations, it showed promise as a therapeutic agent for autoimmune diseases. While we acknowledge possible off-target effects of ERG240 usage in vivo, the specificity towards Bcat1 and

**Table 1 | Pharmacokinetics parameters determined after IV administration of 125 mg kg$^{-1}$ of ERG240 in mice.**

| $C_0$ (µg ml$^{-1}$) | $t_{1/2}$ (h) | AUC$_{last}$ (h × µg ml$^{-1}$) | AUC$_{inf}$ (h × µg ml$^{-1}$) | CL (ml min$^{-1}$ kg$^{-1}$) | MRT (h) |
|---|---|---|---|---|---|
| 231 | 0.12 | 149 | 149 | 45.6 | 0.55 |

$C_0$, drug concentration at time 0; $t_{1/2}$, drug elimination half-life; AUC$_{last}$, the area under the curve up to the last measurable concentration; AUC$_{inf}$, the area under the curve extrapolated to infinity; CL: clearance rate; MRT, mean residence time.

**Table 2 | Pharmacokinetics parameters determined after oral administration of 500 mg kg$^{-1}$ of ERG240 in mice.**

| $C_{max}$ (µg ml$^{-1}$) | $T_{max}$ (h) | AUC$_{last}$ (h × µg ml$^{-1}$) | AUC$_{inf}$ (h × µg ml$^{-1}$) | MRT (h) | F (%) |
|---|---|---|---|---|---|
| 351 | 1.0 | 794 | 794 | 1.24 | 100 |

$C_{max}$, maximum drug concentration in plasma; $t_{max}$, time to $C_{max}$; F, bioavailability.

**Figure 4 | Bcat1 inhibition blocks itaconate production in vivo and polarizes peritoneal macrophages.** Mouse peritoneal macrophages from vehicle (saline), LPS-injected (1.5 mg kg$^{-1}$) or LPS + ERG240 (500 mg kg$^{-1}$)–injected mice were isolated and pooled 24 h following i.p. injection (n = 6 mice per group). Irg1 protein (**a**), mRNA (**b**) levels, as well as itaconate measurements by GC/MS (**c**) are shown. The lower panel (**c**) shows itaconate levels in peritoneal macrophages isolated from mice injected either with LPS or LPS + ERG240 or a sequential treatment where LPS was first injected for 3 h, followed by ERG240 (LPS + ERG240 (sequential)). Peritoneal macrophages were collected 24 h following the LPS injection. n = 3 mice/group. (**d**) Protein–protein interaction (PPI) network in 100 top (97 annotated) differentially expressed genes between LPS and LPS + ERG240 treated mouse peritoneal macrophages illustrated by STRING (high confidence score = 0.7). The IFN-inducible GTPase cluster is shown within a dashed circle. (**e**) Distribution of the fold changes and the FDR-adjusted P values ($-\log_{10}$ (FDR)) for the comparison between LPS and LPS + ERG240 treated macrophages. The IFN- inducible GTPase genes, M1-like, M2-like and extracellular matrix transcripts are shown in blue. (**f**) Fragments per kilobase of transcript per Million mapped reads (FPKM) values for Nos2, Il6 are shown in vehicle, LPS and LPS + ERG240-treated peritoneal macrophages. (**g**) FPKM values for Ifit1, Ifit2, Ifit3, Gbp1, Gbp2, Gbp4, Gbp7, Mrc1, Klf4, Cd36, Fn1, Mmp9, Gsn and Spp1. FPKM = 1 is shown as a dashed line to denote minimal expression levels. All transcripts analysed by RNA-seq (**f,g**) show significant FPKM upon ERG240 treatment when compared to LPS group (FDR < 10$^{-73}$). Error bars are s.e.m. Significance was tested using one-way ANOVA. *$P < 0.05$ by ANOVA.

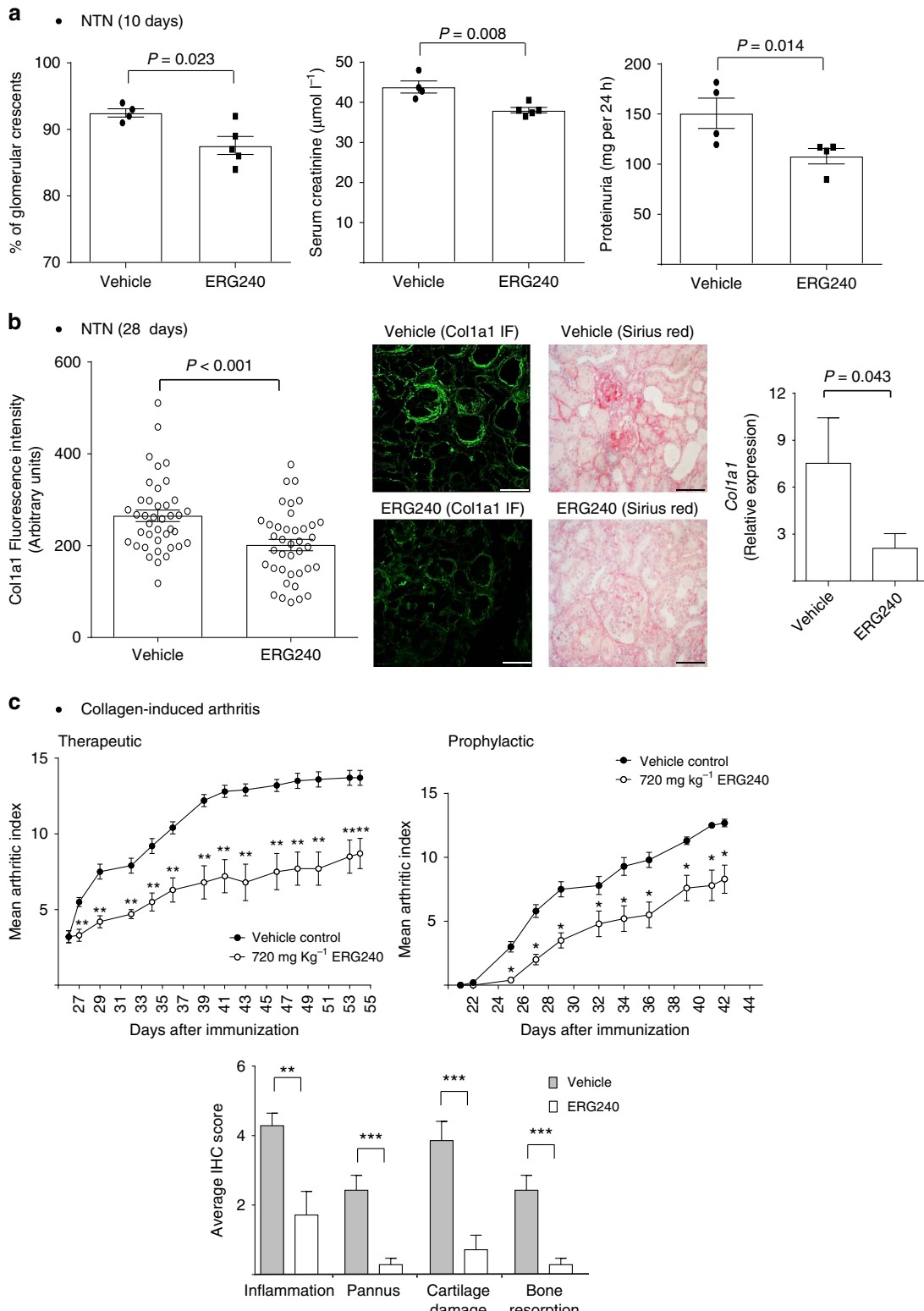

**Figure 5 | Orally administered ERG240 reduced the severity of crescentic glomerulonephritis in rats and CIA in mice.** (**a**) Glomerular crescents, serum creatinine, and proteinuria levels measured in vehicle and ERG240 treated rats at day 10 following induction of nephrotoxic nephritis. At least $n = 4$ rats were used in each group. (**b**) Col1a1 immunofluorescence (IF) quantification (left panel), representative IF and Sirius red staining images (middle panel) and *Col1a1* qRT-PCR (normalized to *Hprt*) 28 days following NTS injection. Original magnification, × 20, at least $n = 4$ animals were used in each group. (**c**) Clinical progression of CIA measured by the Mean Arthritis Index in vehicle and ERG240-treated animals in both studies (prophylactic and therapeutic, upper panel; at least $n = 6$ animals per group). Histological analysis of joints from representative whole paws scored for inflammation, pannus formation, cartilage damage and bone resorption (lower panel). Error bars are s.e.m. Either Mann–Whitney or Student's *t*-test were used for significance. *** $P < 0.001$, ** $P < 0.01$, * $P < 0.05$. Scale bars, 50 μm.

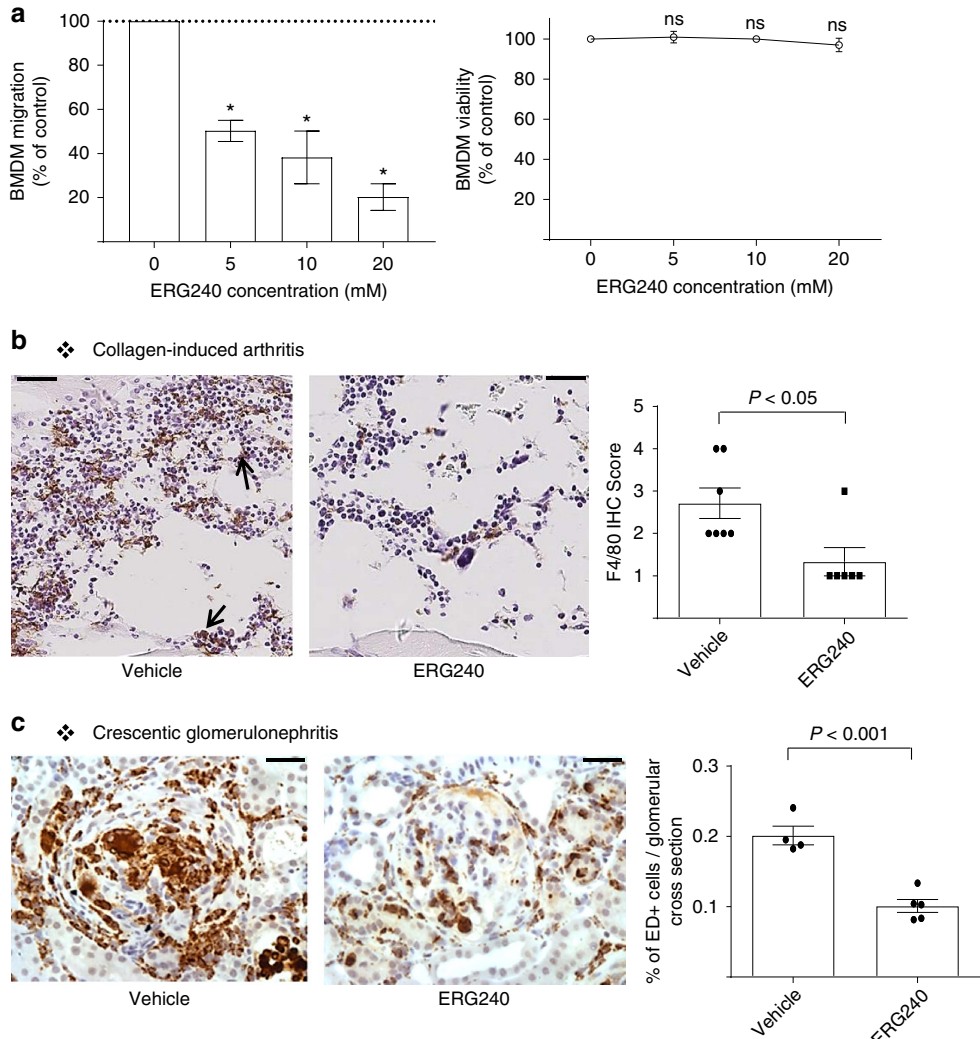

**Figure 6 | ERG240 treatment results in reduced macrophage infiltration in CIA and crescentic glomerulonephritis.** (**a**) Inhibition of murine BMDM migration *in vitro* in the presence of different concentrations of ERG240 by transwell migration assay (left panel, results are presented as the average of four experiments). The drug had no effect on cell viability (right panel, results are presented as the average of three experiments). (**b**) IHC analysis of CIA joints for expression of F4/80. Representative images show F4/80 + cells (indicated with an arrow). F4/80 + cell quantification is shown on the right panel. (**c**) Representative images showing glomerular ED1 (rat CD68)-positive cells in ERG240 treated animals following NTN. Percentage of glomerular ED1-positive cell quantification is shown on the right panel. Error bars are s.e.m. Either one-way ANOVA or Student's *t*-test were used for significance. *$P < 0.001$; ns, non-significant when compared with 0 mM ERG240 by one-way ANOVA. Scale bars, 50 µm (collagen-induced arthritis); 20 µm (crescentic glomerulonephritis).

not its mitochondrial isoform (Bcat2) is supported by the contrasting physiological consequences observed as a result of disrupting *Bcat1* and *Bcat2* genes, individually. $Bcat2^{-/-}$ mice show 50-fold elevation in plasma BCAA concentrations, which leads to organ hypertrophy[63], seizures and death (unpublished observation) if the animals are not offered a low-branched-chain amino acid diet[64]. In contrast, $Bcat1^{-/-}$ mice have normal plasma BCAA concentrations when fed a standard rodent chow[38], suggesting that if there were any non-specific pharmacological effects of ERG240 towards Bcat2 *in vivo*, this would possibly lead to notable pathological consequences which were not observed in neither of the disease models in rats and mice.

In summary, we show that blocking Bcat1 results in therapeutic outcome in two independent models of autoimmune diseases with end-organ damage (joints and glomeruli). We show that macrophages express relatively high amounts of Bcat1 and blocking its activity inhibit Irg1/itaconate levels in human

macrophages, and *in vivo*. Furthermore, Bcat1 inhibition is also associated with reduced macrophage migration *in vitro* and decreased tissue infiltration in arthritis and glomerulonephritis. Thus, macrophage BCAT1 and its ability to interfere with metabolic reprogramming is an attractive pharmacological target for the treatment of chronic inflammatory diseases.

## Methods

**Animals.** DBA/1 mice (male, 6–8 weeks old) used in CIA experiments were purchased from Harlan Sprague Dawley Inc. WKY (male, 16-weeks old, WKY/NCrl) rats used for NTN experiments were purchased from Charles River UK. All mice and rats were used straight from the source by housing them until the appropriate experimental age. All procedures were performed in accordance to institutional guidelines and procedures approved by the UK Home Office (United Kingdom Animals Scientific ProceduresAct, 1986) or followed the requirements of the United States of America Animal Welfare Act (1966) and the guidelines outlined in the Guide for the Care and Use of Laboratory Animals, 8th Edition (2011).

**Cells and reagents.** Human monocyte-derived macrophages were differentiated from buffy coats from healthy donors using gradient separation (Histopaque 1077, Sigma) and adhesion purification. Following Histopaque separation, peripheral blood mononuclear cells were re-suspended in RPMI (Life Technologies) and monocytes were purified by adherence for 1 h at 37 °C, 5% $CO_2$. The monolayer was washed 6 times with HBSS to remove non-adherent cells and monocytes were matured for 5 days in RPMI containing 100 ng ml$^{-1}$ M-CSF (PeproTech, UK) and 10% Fetal Calf Serum (FCS, Labtech International). Macrophage purity was confirmed by immunohistochemical assessment of CD68 and >99% cells were CD68$^+$. For migration assays, C57BL/6 mouse bone marrow derived macrophages (BMDMs) were purchased from Cell Biologics. The cells were cultured in culture media according to manufacturer's instructions. The phenotype of the BMDMs was verified by determining the expression of macrophage marker CD68 by whole cell ELISA. The migration assay is detailed in Supplementary Methods. For BCAA deprivation assay, RPMI 1640 containing L-Glutamine but no BCAAs was used.

Lipopolysaccharide (LPS; *Escherichia coli* serotype 0111:B4; Cat No L4391) was purchased from Sigma (LPS; *Escherichia coli* serotype 0111:B4; Cat No L4391) and the antibodies used in Western Blot analyses are as follows: IL-1β (Cell Signaling Technologies, #12703; D3U3E, dilution 1/1,000), IRG1 (Abcam, ab122624, rabbit polyclonal, dilution, 1/1,000), HIF-1α (NOVUS biological, NB100-449, rabbit polyclonal, dilution 1/2,000), ACTB (Santa Cruz, sc-47778, β-Actin (C4), mouse monoclonal, dilution 1/2,000), BCAT1 (anti Bcat1 raised in rabbit, produced in Susan Hutson's laboratory[4,65], 8382, dilution 1/120,000), TUBB (Cell Signaling Technologies, #2128, rabbit monoclonal, dilution 1/30,000).

**Western blot and RNA analysis.** hMDMs were lysed in Laemmli sample buffer supplemented with protease inhibitors and resolved by SDS-PAGE, transferred into PVDF membranes, and subjected to immunoblotting with the primary antibodies described above and secondary detection antibodies. The probed proteins were detected using SuperSignal West Femto Chemiluminescent Substrate (Thermo Fisher Scientific Inc., Rockford, IL). Uncropped scans of western blots presented in the main and Supplementary Figures are provided in Supplementary Fig. 8. qRT-PCR reactions were performed using the Viaa 7 Real-Time PCR system (Life technologies). A two-step protocol was used beginning with cDNA synthesis with iScript select (Bio-Rad) followed by PCR using Brilliant II SYBR Green QPCR Master Mix (Agilent). A total of 10 ng of cDNA per sample was used. Viia 7 RUO Software was used for the determination of Ct values. Results were analysed using the comparative Ct method and each sample was normalized to the indicated reference gene, to account for any cDNA loading differences. For RNAi, hMDMs were re-plated in six-well plates ($1 \times 10^6$ cells per well) in RPMI (Invitrogen) overnight and transfected with siGENOME SMARTpool for human *BCAT1* (100 nM, Dharmacon SMART pool) or non-targeting siRNA pool as the scrambled control siRNA using Dharmafect 1 (1:50, Dharmacon) as a transfection reagent in OPTIMEM medium (Invitrogen). Following 8 h incubation with OPTIMEM media containing either *BCAT1* or non-targeting siRNA, the culture media was washed and cells were further cultured for 48 h in presence of RPMI media containing M-CSF and FCS. Cells were then used either for extracellular flux analysis (see below) or IRG1, BCAT1, ACTB and TUBB Western Blot analysis. Primer sequences for siRNA and qRT-PCR are provided in Supplementary Fig. 7.

**Metabolism assays.** Real-time measurements of OCR and ECAR were performed using a Seahorse XF96 Extracellular Flux Analyzer (Seahorse Bioscience). hMDMs cultured for 5 days in presence of RPMI containing M-CSF (100 ng ml$^{-1}$) and FCS (10%) were washed and incubated with either ERG240 or LPS or a combination of both for 3 h. Cells were then washed, re-suspended using the non-enzymatic cell dissociation buffer (Sigma) and $5 \times 10^5$ hMDMs were seeded as a monolayer in a 96-well microplate containing XF Assay Modified DMEM. Glucose and different metabolic drugs were injected during real-time measurements of OCR and ECAR. Basal respiration was calculated as the last measurement before addition of oligomycin – non mitochondrial respiration (minimum rate measurement after Rot/AntA). Estimated ATP production designates the last measurement before addition of oligomycin – minimum rate after oligomycin. Maximal respiration is shown as the maximum rate measurement after addition of FCCP – non mitochondrial respiration (minimum rate measurement after Rot/AntA). Glycolysis refers to ECAR values before the addition of oligomycin – ECAR values before the addition of glucose. Glycolytic capacity shows ECAR values following addition of oligomycin – ECAR values before the addition of glucose. In RNAi experiments, hMDMs transfected with either *BCAT1* siRNA or control (non-targeting) siRNA were seeded in a 96-well microplate, LPS-stimulated (100 ng ml$^{-1}$, 3 h) and real-time measurements of OCR and ECAR were recorded, as described above.

**Mouse i.p. injection of LPS and ERG240.** The procedure was performed by Washington Biotechnology Inc. (Baltimore, MD). 6–8 week old female C57BL6 mice were administered ERG240 (500 mg kg$^{-1}$) by intraperitoneal route 30 min before the i.p. LPS injection (from *E. coli* strain 055:B5, 1.5 mg kg$^{-1}$). Same dose of ERG240 was administered a second time, 8 h following the first injection of LPS. Peritoneal macrophages were harvested 24 h after the LPS injection and results

from three groups (vehicle, LPS only, LPS + ERG240; $n = 6$ mice in each group) were analysed in terms of IRG1 and itaconate level measurements.

**Macrophage itaconate and citrate measurements by GC/MS.** For itaconate, cells pellets containing $1.5 \times 10^6$ hMDMs were quenched with 400 μl of methanol, previously chilled at $-20$ °C. After addition of 400 μl of cold (4 °C) ultrapure water, cell pellets were vortexed briefly and then allowed to stand for 10 min on ice. Following addition of 400 μl of GC/MS grade dichloromethane, previously chilled at $-20$ °C, samples were vortexed again and centrifuged at 14,000$g$ for 10 min at 4 °C. The aqueous layer of the samples was collected and lyophilized. Dried materials were dissolved in 40 μl of MOX solution, which is prepared by allowing 20 mg of methoxamine HCl (Sigma Aldrich) to dissolve in 500 μl pyridine for 30 min at room temperature. Metabolites dissolved in MOX were incubated at 45 °C for 60 min, derivatized by adding 60 μl of N-methyl-N-(trimethylsilyl) trifluoroacetamide (MSFTA, Mecherey-Nagel) and incubated at 45 °C for 30 min. After a brief centrifugation, derivatized metabolites were transferred into clean GC/MS vials and analysed for the presence of itaconic acid (itaconate) using a 7890A GC System (Agilent Technologies) equipped with an Agilent J&W DB-5ms column (30 m, 0.25 mm, 0.25 μm), connected to an 5975C Triple-Axis MS Detector (Agilent Technologies) and outfitted with an autosampler. The MS source was held at 230 °C and the quadrupole at 150 °C. The GC oven temperature was held at 50 °C for 5 min, increased to 240 °C at a rate of 10 °C min$^{-1}$, held at 240 °C for 5 min and increased to 300 °C at a rate of 10 °C min$^{-1}$. A volume of 1 μl was injected in the sample in a 1:3 split mode. The amount of itaconate present in the sample was determined after peak integration. For citrate, the frozen cell pellets were re-suspended in ice-cold ethanol-phosphate buffer. Cells were lysed by three freeze-thaw cycles and sonication. Samples were centrifuged and the supernatant was used for further analysis. The GC/MS was performed by Biocrates (BIOCRATES Life Sciences AG, Innsbruck, Austria) and the citrate quantification was performed using their commercially available Kit plates, according to the manufacturer's instructions. After derivatization to its corresponding methoxime-trimethylsilyl (MeOx-TMS) derivative, citrate levels were determined by GC/MS using Agilent 7890 GC/5975 MSD (Agilent, Santa Clara, USA). Pretreated samples were evaporated to complete dryness and subjected to a two-step methoximation-silylation derivatization. N-methyl-N-(trimethylsilyl) trifluoroacetamide (MSTFA) was used as silylation reagent. Split injection was performed and chromatograms were recorded in selected ion monitoring mode. External standard calibration curves and ten internal standards were used to calculate concentrations of individual energy metabolites. Data were quantified using the appropriate MS software (Agilent, Masshunter) and imported into Biocrates MetIDQ software for further analysis.

**In vivo pharmacokinetics.** The experiment was performed by Eurofins Panlabs Inc. Briefly, fasting pharmacokinetic parameters were obtained after administration of 125 mg kg$^{-1}$ i.v. or 500 mg kg$^{-1}$ p.o ERG240 to ICR mice and blood collection at 3, 10, 30, 60, 120 and 240 min (i.v. dosing) or 10, 30, 60, 120, 240 and 360 min (p.o. dosing). Each animal was subjected to a single blood draw and 3 animals per time point were used. Parameters were obtained from non-compartmental analysis of the plasma data using WinNonlin. Experimental details are provided in Supplementary Methods.

**RNA-sequencing.** Total RNA was extracted from hMDMs and mouse peritoneal macrophages using Trizol (Invitrogen) according to manufacturer's instructions with an additional purification step by on-column DNase treatment using the RNase-free DNase Kit (Qiagen) to ensure elimination of any genomic DNA. The integrity and quantity of total RNA was determined using a NanoDrop 1000 spectrophotometer (Thermo Fisher Scientific) and Agilent 2100 Bioanalyzer (Agilent Technologies). In total 500 ng of total RNA was used to generate RNA-seq libraries using TruSeq RNA sample preparation kit (Illumina) according to the manufacturer's instructions (see Supplementary Methods for detailed library preparation, mapping and sequencing protocols).

**Collagen-induced arthritis in mice.** DBA/1 mice (6–8 weeks old from Harlan Sprague Dawley Inc.) were injected s.c. with 100 μg of bovine type II collagen emulsified in complete Freund's adjuvant (Sigma). At day 21 following primary immunization, the mice were given a booster injection of 100 μg of bovine type II collagen emulsified in incomplete Freund's adjuvant. The prophylactic arm of the study involved water-treated (vehicle control, $n = 6$) and ERG240-treated (720 mg kg$^{-1}$, o.p., qdx5, 3 weeks, $n = 10$) animals, while the therapeutic arm of the study involved water-treated (vehicle control, $n = 10$) and ERG240-treated (1,000 mg kg$^{-1}$, o.p., qdx5, 4 weeks, $n = 10$) animals. Parafin-embedded paws were sectioned at ~5 μm. One slide from each paw was stained with hematoxylin and eosin (H&E) and served as a reference, while four additional slides were stained using antibodies against F4/80 (AbD Serotec). The expression of the above markers in the joints of the preserved CIA tissues was scored according to the following scale: 0 = no staining; 1 = minimal; 2 = mild; 3 = moderate; 4 = marked; and 5 = diffuse and severe staining (see Supplementary Methods for detailed histological evaluation of CIA tissues). Serum levels of TNF and RANKL were analysed using relevant ELISA kits from Biolegend (TNF-α) and Abcam (RANKL)

according to the manufacturers' instructions. Details for determining the presence of antibodies against type II collagen are provided in Supplementary Methods.

**Nephrotoxic nephritis in the WKY rat.** Nephrotoxic nephritis (NTN) was induced in twelve week old male Wistar-Kyoto (WKY) rats by intravenous injection of 0.1 ml of nephrotoxic serum (NTS), as previously described[66]. Five WKY rats received ERG240 (250 mg kg$^{-1}$) orally in 0.1% carboxymethylcellulose once a day for a duration of 10 days whereas the vehicle treated WKY rats had carboxymethylcellulose only for the same period. Nine days following NTS injection, urine was collected by placing rats in metabolic cages for 24 h with free access to food and water. Proteinuria was determined by the sulphosalicylic acid test and serum creatinine by the Creatinine Assay Kit (Abcam). After 10 or 28 days of NTN induction, rats were culled and the kidney was formalin fixed and paraffin embedded for immunohistochemistry. To quantify the number of macrophages infiltrating the glomeruli, formalin-fixed paraffin-embedded kidney sections were stained with mouse monoclonal antibody to ED-1 (Serotec, Oxford, UK), followed by an HRP-labelled anti-mouse polymer development system (Dako Ltd, UK). For the evaluation of total collagen, renal tissues were stained with Sirius red. For Col1a1 immunofluorescence, sections were blocked with BSA, incubated with the Col1a1 primary antibody raised in goat (Southern Biotech, 1/40 dilution), washed and re-incubated with secondary antibody (donkey anti-goat-FITC, 1/100 dilution, Abcam). For ED-1 staining, the cellular infiltrates in 20 consecutive glomeruli were quantified using an automated image analysis software (ImagePro Plus, Media Cybernetics, Bethesda, MD) and expressed as a percentage of total glomerular cross sectional area.

**Statistical analyses.** Data are presented as mean ± s.e.m. and analysed using GraphPad Prism software (version 7.02; GraphPad). The Mann–Whitney or Student's $t$-tests were used for comparison of two groups. Differences in percentage of itaconate in human samples were tested for significance using a one-sample-$t$-test. One-way or two-way ANOVA analyses (followed by Tukey's or Dunnet's or Sidak's multiple comparison tests) were used for comparative analyses of three or more groups.

**Data availability.** Mouse sequence data that support the findings of this study have been deposited in the European Nucleotide Archive with the primary accession code PRJEB20215. The authors declare that all other data supporting the findings of this study are available within the paper and its Supplementary Information files or are available from the corresponding authors on request.

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

## Acknowledgements

This work was supported by the Medical Research Council (MR/M004716/1 and MR/N01121X/1 to J.B.) and by Kidney Research UK (RP9/2013 to J.B.). D.C. is supported by the Institute Pasteur, Fondazione Cenci Bolognetti. C.M. is supported by the British Hearth Foundation Fellowship (FS/12/3829640). We are grateful to Dr Susan Hutson (Virginia Tech) for the generous gift of BCAT1 antibody, which was used to detect BCAT1 protein in human macrophages by E.A.A.

## Author contributions

A.E.P., C.M. and J.B. designed the experiments. J.-H.K. performed the glomerulonephritis experiments together with Western Blot, qRT-PCR assays. E.A.A. performed BCAT1 Western Blot analysis. M.I. and M.B. generated the RNA-seq and analysed the human RNA-seq data with J.B. M.B. has generated the mice RNA-seq data, which was analysed by P.K.S. and J.B. S.P.M. measured the serum creatinine levels. A.E.P. and H.A.V. generated ERG240 and performed and analysed all pharmacological and *in vivo* CIA assays. A.E.P. measured itaconate levels by GC/MS. D.C. and C.M. performed the extracellular flux analyses and analysed the data. J.B. oversaw the study and wrote the manuscript with contributions from A.E.P., C.M. and E.A.A. All authors discussed and approved the results presented in the manuscript.

## Additional information

**Competing interests:** A.E.P. is an employee of Ergon Pharmaceuticals Ltd. The remaining authors declare no competing financial interests.

