## [Peer Review File · Nature Communications]

Reviewers' comments:

Reviewer #1 (Remarks to the Author):

Papathanassiou et al. present a straightforward study using a novel BCAT1 inhibitor, complemented by siRNA for BCAT1 using human monocyte-derived macrophages as well as murine and rat models. Authors report that BCAT1 is primary transporter in macrophages. They have developed a novel small molecule inhibitor to BCAT1 called ERG240. Inhibition decreases IRG1 and synthesis of itaconate, as well as decreases bioenergetics. Further, they showed that inhibition of BCAT1 reduced pro-inflammatory transcriptome signature- RNAseq analysis of peritoneal macrophages show that the drug ablates several pro-inflammatory genes upregulated by LPS, while increasing immunomodulatory (or alternative-like) macrophage markers. The data are clear that the drug blocks two autoimmune models of disease in a rat and murine model, albeit at a high dose, through blunting of macrophage infiltration and or metabolism. This work is exciting because little work has been completed on BCAA metabolism in macrophage biology. This paper proposes an exciting novel drug and target in metabolic reprogramming of macrophages that will have broad interest in the field. What is less clear is the mechanism of how BCAT1 inhibition regulates disease outcome- the inhibitor blunts both macrophage infiltration and metabolism/inflammation.

Several main concerns are presented that can be addressed through edits or additional experiments that will strengthen the manuscript:

1- In vitro LPS + drug studies shown are in designs where the cells are co-treated (LPS+ drug at same time). This does not address if the drug can reverse existing LPS-induced inflammation, which would be more appropriate to therapeutic approaches. Adding an in vitro cell culture experiment showing if the drug can reverse inflammation would contribute to understanding if in vivo findings are simply due to decreased macrophage infiltration or intrinsic macrophage changes in metabolism.

2- Several concerns about the interpretation of the metabolic assays should be addressed. The inhibitor appears to inhibit both ECAR and OCR for Seahorse bioenergetic studies such as glycolytic rate, etc. (although no stats are presented, and should be). This means that both oxygen consumption (OCR) and glycolysis (ECAR) are reduced by the inhibitor. First, authors mention that oxidative phosphorylation is inhibited but it should be discussed that in macrophages, NADPH oxidase can also use oxygen in the "oxidative burst", along with the electron transport chain (oxphos). Therefore, saying just "oxidative phosphorylation" is potentially misleading. Authors should say "oxygen consumption" because that is all that the Seahorse measures unless further studies are conducted to specifically show that it is specifically inhibition of oxphos. Second, importantly, siRNA knockdown of BCAT1 appears to only reduce OCR, not ECAR- thus the drug is inhibiting glycolysis while knockdown of BCAT1 by siRNA does not decrease ECAR/glycolysis. Studies should be repeated in cells where BCAT1 has been knocked down to determine specificity for BCAT1 inhibitor since it appears that the drug is doing more than the knockdown. Third, authors state the importance of BCAT1 in CD4+ T cells regulating glycolysis, but fail to discuss their findings on glycolysis (ECAR) here. Typically, glycolytic cells are associated with pro-inflammation while inhibition of mitochondrial metabolism (Vats/Chawla Cell Met and others) show that the "M2/alternative-like" phenotype of macrophage is dependent upon mitochondrial metabolism. The role of BCAT1 in glycolysis as a potential mechanism must be discussed. Fourth, if the drug is blocking ECAR and OCR, how long can the cells live?

Minor:

Add additional explanation for "extracellular space" findings because this is unclear.

Authors fail to cite and discuss this paper found in quick google search "Nemeth, B.; Doczi, J.; Csete, D.; Kacso, G.; Ravasz, D.; Adams, D.; Kiss, G.; Nagy, A. M.; Horvath, G.; Tretter, L.; Mocsai, A.; Csepanyi-Komi, R.; Iordanov, I.; Adam-Vizi, V.; Chinopoulos, C. (2015). "Abolition of mitochondrial substrate-level phosphorylation by itaconic acid produced by LPS-induced Irg1 expression in cells of murine macrophage lineage". The FASEB Journal. doi:10.1096/fj.15-279398. PMID 26358042."

Statistical analysis (or indications in figures) is lacking in Fig 1C. Fig 4C.

Dosing in vitro should be discussed further- 20mM in vitro is a high dose if IC50 is in nM range? Why 20mM?

ATP production should be marked as "estimated ATP production". The Y axis shows that it is a calculation from OCR experiment- not an actual measure of ATP. This is misleading.

A cartoon of metabolites regulated would aid the reader.

Add a citation for "as the public datasets for human immune cell expression indicate,"

Reviewer #2 (Remarks to the Author):

In their manuscript, Papathanassiou et al. describe the role of BCAT1 during inflammation in macrophages. They show that BCAT1 activity is directly linked to the inflammatory response of macrophages and that loss of its activity reduces the inflammatory response determined based on synthesis of itaconic acid (a pro-inflammatory metabolite) and the expression of IRG1. Moreover, the authors describe an inhibitor, ERG240, that seems to specifically inhibit BCAT1 and not BCAT2. Experiments with ERG240 in mouse models of auto-immune diseases highlight the power of this compound as potential drug. The manuscript is of high quality and clearly written. The addressed topic is of high impact, novel and will be of importance to the field. However, there are a few concerns that should be addressed before publication.

1. To get a better understanding on the activation of macrophages it would be essential to see expression levels of other cytokines and markers such as TNF α , IL6 and iNOS.
2. Fig 1b: Based on matrix effects and peptide type, ionization efficiency can vary. Without proper (isotope labeled) control peptides, the abundances of BCAT1 and BCAT2 peptides can not be compared. Maybe a Western blot would do better.
3. The applied concentration of the inhibitor is very high (20 mM). The authors should demonstrate that the inhibitor does not impair cell viability. In addition, they should provide evidence that the inhibitor is not metabolized by the cells and this way causing secondary effects.
4. Fig 2D: BCAT1 expression with non-targeting and targeting siRNA is missing for LPS treatment.
5. Fig 2E and 3C: itaconate levels should be replicated in more than one experiment and error bars be added.
6. It would be interesting to know why the half-life in animals is very low. Again, is the drug metabolized by the organism? (see also point 3).
7. Leucine is degraded to acetyl-Coa and acetoacetate both fueling cellular acetyl-Coa pools. Does decreased activity of BCAT1 impact cellular acetyl-Coa and citrate levels? Itaconate synthesis is a sink for citrate and diverts carbon away from the TCA cycle. It would be of importance to see whether acetyl-Coa levels are affected by BCAT1 inhibition and if this translates through to the observed citrate levels.
8. One option to test whether the function of BCAT1 is independent of its transaminase activity would be to deprive cells of BCAA and see if the effect compares to BCAT1 silencing or inhibition through ERG240.

Reviewer #1 (Remarks to the Author):

Papathanassiou et al. present a straightforward study using a novel BCAT1 inhibitor, complemented by siRNA for BCAT using human monocyte-derived macs as well as murine and rat models. Authors report that BCAT1 is primary transporter in macs. They have developed a novel small molecule inhibitor to BCAT1 called ERG240. Inhibition decreases IRG1 and synthesis of itaconate, as well as decreases bioenergetics. Further, they showed that inhibition of BCAT1 reduced pro-inflammatory transcriptome signature- RNAseq analysis of peritoneal macs show that the drug ablates several pro-inflammatory genes upregulated by LPS, while increasing immunomodulatory (or alternative-like) macrophage markers. The data are clear that the drug blocks two autoimmune models of disease in a rat and murine model, albeit at a high dose, through blunting of macrophage infiltration and or metabolism. This work is exciting because little work has been completed on BCAA metabolism in macrophage biology. This paper proposes an exciting novel drug and target in metabolic reprogramming of macrophages that will have broad interest in the field. What is less clear is the mechanism of how BCAT1 inhibition regulates disease outcome- the inhibitor blunts both macrophage infiltration and metabolism/inflammation.

Several main concerns are presented that can be addressed through edits or additional experiments that will strengthen the manuscript:

1. *In vitro* LPS + drug studies shown are in designs where the cells are co-treated (LPS+ drug at same time). This does not address if the drug can reverse existing LPS-induced inflammation, which would be more appropriate to therapeutic approaches. Adding an *in vitro* cell culture experiment showing if the drug can reverse inflammation would contribute to understanding if *in vivo* findings are simply due to decreased macrophage infiltration or intrinsic macrophage changes in metabolism.

Authors' reply. We thank the reviewer for the thorough and accurate review of the findings. Since this is a crucial point, we performed an *in vivo* experiment where mice were injected with *i*) LPS, *ii*) LPS and ERG240 at the same time or *iii*) LPS for 3 hours, followed by ERG240, and peritoneal macrophages were collected 24 hours after LPS injection. Itaconate levels as well as *Il6*, *Gbp2*, *Gbp4*, expression levels (representative transcripts of IFN-induced GTPases and M1-like activation) were measured and presented below.

Itaconate levels and *Il6*, *Gbp2*, *Gbp4* mRNA expression in peritoneal macrophages isolated from mice injected either with LPS or LPS + ERG240 or a sequential treatment where LPS was first injected for 3 hours, followed by ERG240 (LPS + ERG240 (sequential)). Peritoneal macrophages were collected 24h following the LPS injection and subjected to GC/MS (itaconate) and qRT-PCR (*Il6*, *Gbp2*, *Gbp4* expression). n=3 mice / group; All expression values are normalized to those obtained for *Actinb* mRNA levels. *, P < 0.05, ***, P < 0.001, when compared with LPS-treated samples.

These results show that ERG240 can still reduce significantly itaconate production when injected after an initial inflammation is established in the mouse peritoneum. In addition, when ERG240 is administered following 3 hours of LPS challenging, there is an inhibitory effect on *Il6*, *Gbp4* expression levels, suggesting that the drug can maintain its anti-inflammatory properties, at least in part. This is also in line with the CIA experiment where ERG240 alleviates the severity of CIA when given orally after arthritis is established (Fig 5C in the revised manuscript, 4C in the submitted manuscript). As suggested by the reviewer, ERG240 is therefore likely to affect the intrinsic changes in macrophage metabolism. As a result, we now include the findings describing the sequential treatment (Itaconate measurements by GC/MS in mice treated with LPS for 3 hours followed by ERG240) in the revised version of the main manuscript (Figure 4C, lower panel).

2. Several concerns about the interpretation of the metabolic assays should be addressed. The inhibitor appears to inhibit both ECAR and OCR for Seahorse bioenergetic studies such as glycolytic rate, etc. (although no stats are presented, and should be). This means that both oxygen consumption (OCR) and glycolysis (ECAR) are reduced by the inhibitor

Authors' reply. Statistics are now added on all ECAR and OCR measurements' results as separate bar graphs and included in the revised manuscript's Figure 2A and B (see also the response to point 4 where statistics are also included for the revised siRNA experiments showing that indeed both ECAR and OCR are reduced).

3. First, authors mention that oxidative phosphorylation is inhibited but it should be discussed that in macrophages, NADPH oxidase can also use oxygen in the "oxidative burst", along with the electron transport chain (oxphos). Therefore, saying just "oxidative phosphorylation" is potentially misleading. Authors should say "oxygen consumption" because that is all that the Seahorse measures unless further studies are conducted to specifically show that it is specifically inhibition of oxphos.

Authors' reply. The reviewer is correct that what was measured is oxygen consumption and the term 'oxidative phosphorylation' has now been replaced by 'oxygen consumption' throughout the revised manuscript (see highlighted text and figure legends).

4. Second, importantly, siRNA knockdown of BCAT1 appears to only reduce OCR, not ECAR- thus the drug is inhibiting glycolysis while knockdown of BCAT1 by siRNA does not decrease ECAR/glycolysis. Studies should be repeated in cells where BCAT1 has been knocked down to determine specificity for BCAT1 inhibitor since it appears that the drug is doing more than the knockdown.

Author's reply. According to the reviewer's suggestion, we have performed additional siRNA experiments in 4 independent donors and reported the effect of *BCAT1* knockdown on OCR and ECAR in LPS-stimulated human macrophages using full Seahorse analysis to allow direct comparison with results presented in Figures 2A and B. The results shown below are now included in the revised manuscript as Figures 2C and D (text, legend and methods are changed accordingly). Statistical analysis on all OCR and ECAR measurements following *BCAT1* knockdown are presented in the revised Figure 2D.

These results show that *BCAT1*-siRNA treatment significantly reduces OCR and ECAR, confirming the results obtained with pharmacological blockade (Figures 2A and B). These results are now updated in the abstract, results and discussion.

5. Third, authors state the importance of *BCAT1* in *CD4+* T cells regulating glycolysis, but fail to discuss their findings on glycolysis (ECAR) here. Typically, glycolytic cells are associated with pro-inflammation while inhibition of mitochondrial metabolism (Vats/Chawla Cell Met and others) show that the “M2/alternative-like” phenotype of macrophage is dependent upon mitochondrial metabolism. The role of *BCAT1* in glycolysis as a potential mechanism must be discussed.

Authors’ reply. The role of *BCAT1* in glycolysis is now discussed by adding the following paragraph (see also highlighted text in the discussion):

Metabolism in M(LPS) macrophages is characterised by high levels of glycolysis and decreased respiration, a fragmented TCA and relatively high levels of some of its intermediates (succinate, citrate and itaconate). By contrast, metabolism in M2 (or M(IL-4)) corresponds to high levels of fatty acid oxidation and oxidative phosphorylation. Our findings indicating (i) the reduced ECAR levels as a result of *BCAT1* inactivation in human macrophages, (ii) the anti-inflammatory effects of ERG240, reversing the M1-like transcriptome in mouse peritoneal macrophages, argue in favour of the involvement of BCAA in M1/pro-inflammatory macrophage function.

6. Fourth, if the drug is blocking ECAR and OCR, how long can the cells live?

Authors’ reply. According to the reviewer’s suggestion, a cell viability assay was performed on human MDMs (Please see also the response to Reviewer 2, point 3). ERG240 did not affect cell viability at 3h and 8h LPS stimulation neither in control or LPS-treated human macrophages. The results are presented in the revised manuscript, as well as in response to Reviewer 2’s point below (Reviewer 2, point 3).

Minor:

Add additional explanation for “extracellular space” findings because this is unclear.

Authors’ reply. ‘Extracellular space’ has now been replaced by ‘extracellular matrix genes’ in the revised manuscript (Figure 3 in the submitted manuscript which is now Figure 4 in the revised one). We have also added a Volcano plot showing the differentially expressed genes (Figure 4E in the revised manuscript).

Authors fail to cite and discuss this paper found in quick google search “Nemeth, B.; Doczi, J.; Csete, D.; Kacso, G.; Ravasz, D.; Adams, D.; Kiss, G.; Nagy, A. M.; Horvath, G.; Tretter, L.; Mocsai, A.; Csepanyi-Komi, R.; Jordanov, I.; Adam-Vizi, V.; Chinopoulos, C. (2015). “Abolition of mitochondrial substrate-level phosphorylation by itaconic acid produced by LPS-induced Irg1 expression in cells of murine macrophage lineage”. The FASEB Journal. Doi:10.1096/fj.15-279398. PMID 26358042.”

Authors’ reply. This paper is now cited and discussed by adding this following paragraph to the discussion (highlighted text).

Besides its antimicrobial effects, itaconate has been proposed as an inhibitor of substrate level phosphorylation (SLP) through its effect on succinate-CoA ligase in RAW-264.7 cells¹.

Statistical analysis (or indications in figures) is lacking in Fig 1C. Fig 4C.

Authors’ reply. Statistical analyses are added for Fig 1C and 4C. Please note that Figure 4C is now Figure 5C in the revised manuscript. (See also highlighted text in figure legends).

Dosing in vitro should be discussed further- 20mM in vitro is a high dose if IC50 is in nM range? Why 20mM?

Authors’ reply. ERG240 is a structural analogue of leucine, designed based on the X-ray crystal structure of BCAT1 and, as the reviewer correctly states, it can inhibit BCAT1 within 0.1-1nM range of IC50. The concentration of 20 mM used in the cellular assays is because of the dependence of ERG240 on cell transporters for its intracellular inhibitory effect and its possible competition with leucine, which is in mM concentration (0.381 mM) in the RPMI media used throughout the study. Analysis of samples of human macrophage lysates using GC/MS revealed that 1×10^6 cells treated with 3.28 mg ERG240 (corresponding to 20 mM of the drug) will uptake about 0.8-3 ng of the compound with an estimated intracellular ERG240 concentration of ~70-260 nM (See also Author’s reply to Point 3 made by Reviewer 2 below). Considering that only a fraction of the intracellular drug level will reach the target enzyme (with the rest associated with the transporting machinery and other cell sites), the estimated intracellular ERG240 levels is in line with the range of concentrations required to inhibit the enzymatic activity of the BCAT1.

ATP production should be marked as “estimated ATP production”. The Y axis shows that it is a calculation from OCR experiment- not an actual measure of ATP. This is misleading.

Authors’ reply. ATP production is now labelled as ‘estimated ATP production’ and Figures 2 and S2 were modified together with their respective legends as well as the results/methods section.

A cartoon of metabolites regulated would aid the reader.

Authors' reply. A cartoon illustrating the flow of metabolites between BCAA and the broken TCA metabolites is now shown in Fig S3.

Add a citation for “as the public datasets for human immune cell expression indicate,”

Authors' reply. The sentence was deleted from the revised manuscript.

Reviewer #2 (Remarks to the Author):

In their manuscript, Papathanassiu et al. describe the role of BCAT1 during inflammation in macrophages. They show that BCAT1 activity is directly linked to the inflammatory response of macrophages and that loss of its activity reduces the inflammatory response determined based on synthesis of itaconic acid (a pro-inflammatory metabolite) and the expression of IRG1. Moreover, the authors describe an inhibitor, ERG240, that seems to specifically inhibits BCAT1 and not BCAT2. Experiments with ERG240 in mouse models of autoimmune diseases highlight the power of this compound as potential drug. The manuscript is of high quality and clearly written. The addressed topic is of high impact, novel and will be of importance to the field. However, there are a few concerns that should be addressed before publication.

1. To get a better understanding on the activation of macrophages it would be essential to see expression levels of other cytokines and markers such as TNF α , IL6 and iNOS.

Authors' reply. We thank the reviewer for his/her careful and constructive review. According to the reviewer's suggestion, we measured the expression of *Il6*, *NOS2*, *TNF* and *PTGS2* as well-established early LPS-response transcripts. The results are presented below and in the revised manuscript (Fig S1-F, 3 hours LPS stimulation; n=6 donors used per group).

2. Fig 1b: Based on matrix effects and peptide type, ionization efficiency can vary. Without proper (isotope labeled) control peptides, the abundances of BCAT1 and BCAT2 peptides cannot be compared. Maybe a Western blot would do better.

Authors' reply. According to the reviewer's suggestion, we have performed Western Blot analyses using BCAT1 and BCAT2 antibodies in hMDMs from 5 separate donors and the results are presented in the figure below. The Western Blot results indicate relatively stronger BCAT1 band intensities when compared with BCAT2 (see below). However because of the different affinities of the two antibodies, a direct quantitative comparison cannot be made, making these results suggestive but not conclusive. Nevertheless, the RNA-seq results showing a relatively increased BCAT1 mRNA copies in 14 healthy donors

backed up by quantitative proteomics by LC-MS/MS (with maximum of the 12 most abundant multiply-charged ions registered in each survey spectrum selected in a data-dependent manner) confirms that increased BCAT1 mRNA levels reflect equally increased protein levels. We follow the reviewer’s advice and withdraw LC-MS/MS results from the revised manuscript.

Western Blot results for BCAT1 and BCAT2 in human MDMs from 5 healthy donors. The samples were stimulated with LPS (100ng/ml, 8 hours) and either treated with ERG240 (20mM, 8h) or left untreated. Following immunoblotting with BCAT1 (upper panel) and BCAT2 (lower panel) antibodies and secondary detection antibodies, the probed proteins were detected after 10s of exposure for both BCAT1 and BCAT2.

3. The applied concentration of the inhibitor is very high (20 mM). The authors should demonstrate that the inhibitor does not impair cell viability. In addition, they should provide evidence that the inhibitor is not metabolized by the cells and this way causing secondary effects.

Authors’ reply. According to the reviewer’s suggestion, we have performed cell viability for ERG240 and/or LPS in hMDMs for the stimulation time points used in the manuscript. We have also included a 24h stimulation time point and the results are now presented below and in the revised Fig S1-E. The manuscript’s results and methods (Supplementary methods, Page 3 highlighted) were modified accordingly.

Cell viability in hMDMs treated with either LPS (100ng/ml) or ERG240 (20mM) or both at 3h, 8h and 24h stimulation periods. n=3 donors were used; ns, non-significant.

The results show that for the stimulation time points used in our study (3h and 8h), there is no significant effect of ERG240 in both control (basal) and LPS-treated primary human macrophages. There was a modest but significant reduction in cell viability at 24h only in the LPS-stimulated cells.

The drug does not undergo any cellular metabolic reaction. ERG240 can easily be detected in lysates of human macrophages by GC/MS. In the graph below, overlapping chromatographs of a standard concentration of 10 ng ERG240 and cell lysates of untreated and ERG-treated human macrophages are shown.

4. *Fig 2D: BCAT1 expression with non-targeting and targeting siRNA is missing for LPS treatment.*

Authors' reply. According to the reviewer's suggestion, we have now included siRNA results for LPS treatment. Figure 2C and D in the submitted manuscript were also changed to take into account Reviewer 1's suggestions (see also the response to Reviewer 1, point 4)

5. *Fig 2E and 3C: itaconate levels should be replicated in more than one experiment and error bars be added.*

Authors' reply. According to the reviewer's suggestion, we have repeated the *BCAT1* siRNA and measurements of itaconate by GC/MS in 4 donors. We have also repeated the *in vivo* LPS + ERG240 injection in mice to have an independent set of biological replicates (Please note that we have also included a group where there was a sequential treatment of LPS and ERG240 according to the reviewer 1's suggestion). The results are shown below and included in the main revised manuscript Figure 2E (which is now Fig 3 in the revised manuscript) and Fig. 3C (which is 4C in the revised manuscript).

Itaconate levels measured by GC/MS were further confirmed in *si-BCAT1* and *si-Control* MDMs from independent donors (n=4, left panel). Itaconate levels in peritoneal macrophages isolated from mice injected either with LPS or LPS +ERG240 or a sequential treatment where LPS was first injected for 3 hours, followed by ERG240 (LPS + ERG240 (sequential)). Peritoneal macrophages were collected 24h following the LPS injection. n=3 mice / group. *, P < 0.05.

6. It would be interesting to know why the half-life in animals is very low. Again, is the drug metabolized by the organism? (see also point 3).

Authors' reply. The pharmacokinetic study that determined the half-life of the drug in mice also showed that the oral bioavailability of the drug was 100%. The latter suggests that ERG240 is completely absorbed and not subject to first-pass metabolism. Since the drug is structurally a leucine analogue, it is likely to be uptaken by the ubiquitously expressed amino acid transporter LAT1. In that light, the low half-life of the drug is related to its complete and swift absorption by various sites rather to a phase I or II metabolic reaction. ERG240 is also capable of crossing the blood brain barrier (BBB) although it does not accumulate preferentially in the brain.

As previously mentioned, the drug does not undergo any metabolic reaction. For the metabolism of the drug in human macrophages, please see the response to point 3, which shows the GC/MS chromatograms with comparative analysis of human macrophages treated with ERG240 (20 mM) and 10 ng of ERG240.

7. Leucine is degraded to acetyl-Coa and acetoacetate both fueling cellular acetyl-Coa pools. Does decreased activity of BCAT1 impact cellular acetyl-Coa and citrate levels? Itaconate synthesis is a sink for citrate and diverts carbon away from the TCA cycle. It would be of importance to see whether acetyl-Coa levels are effected by BCAT1 inhibition and if this translates through to the observed citrate levels.

Authors' reply. These are very relevant suggestions and following the reviewer's advice, we have measured citrate levels in 4 donors following LPS and LPS + ERG240 treatment for 3 hours by GC-MS. We present the results in revised Fig S3 (which replaces the colorimetric assay).

Citrate levels were measured by GC-MS in basal-, ERG240-, LPS-, and LPS+ERG240 treated hMDMs (n=4 donors). LPS treatment was for 3 hours at 100ng/ml. ANOVA followed by Tukey’s multiple comparison test was used for statistical comparisons.

The results show that there is relative accumulation of citrate in LPS+ERG240-treated human macrophages when compared to LPS only. This is in line with the inhibitory role of ERG240 on *IRG1*, which converts cis-aconitate derived from citrate into itaconic acid (See also Fig S3-B for a representative cartoon of metabolites). Despite our efforts, we failed to detect Acetyl-coA in human macrophages by GC/MS, probably due to very instable properties of this metabolite. Nevertheless, the effect of BCAT1 inhibition on the broken TCA cycle in human macrophages lies predominantly downstream citrate at the *IRG1* site with a relatively minor effect through inhibition of BCAA transamination and reduction in the amount of acetyl-coA entering the cycle (Fig S3B).

8. One option to test whether the function of BCAT1 is independent of its transaminase activity would be to deprive cells of BCAA and see if the effect compares to BCAT1 silencing or inhibition through ERG240.

Authors’ reply. According to the reviewer’s suggestion, we have performed an experiment where human macrophages isolated from healthy donors were stimulated with LPS in media deprived from BCAA, followed by measurement of *IRG1* expression and itaconate levels. The results are shown below and are included in the revised manuscript. (Fig S1-C and D).

Human monocyte-derived macrophages were either incubated with standard conditioned media (Standard) or BCAA-deprived for 8 hours in either basal or LPS-treated (100 ng/ml) conditions and *IRG1* expression and itaconate levels were measured by qRT-PCR (left panel) and GC/MS (right panel). Expression values are normalised to *B2M*, n=3 donors, ***, P < 0.001 when compared with LPS (Standard).

As the reviewer predicted, the results are similar to pharmacological inhibition by ERG240 and the silencing experiments' findings. This also suggests that BCAT1 must be enzymatically active to have its effect on IRG1-mediated itaconate production in human macrophages. We thank the reviewer for suggesting this experiment, which added significantly to the understanding of our findings.

Reference

1. Nemeth B, *et al.* Abolition of mitochondrial substrate-level phosphorylation by itaconic acid produced by LPS-induced Irg1 expression in cells of murine macrophage lineage. *FASEB journal : official publication of the Federation of American Societies for Experimental Biology* **30**, 286-300 (2016).

REVIEWERS' COMMENTS:

Reviewer #1 (Remarks to the Author):

Great job at addressing concerns.

Reviewer #2 (Remarks to the Author):

With their revision the authors addressed all points raised by me. In addition they updated the discussion and it is interesting to know that BCAT1 loss of function also impacts Irg1 in BCAA deprived medium suggesting a secondary activity of BCAT1 despite its transaminase activity.

Only two minor points left from this reviewer:

1. On page 7, line 158 of the revised manuscript and also in the initial submission, the authors show increased OCR induced by LPS. However, in the discussion (page 14, line 293) they state that respiration is decreased in M(LPS) macrophages contradicting their own measurements. Most likely, the discussion relates to Palsson-McDermott et al. or Tannahil et al. who observed decreased respiration in LPS activated macrophages. ERG240 reduces OCR.

2. Other functions of itaconate: There are also studies out demonstrating that itaconic acid inhibits SDH (1960s) in general and more recently specifically in macrophages under inflammation (Booth et al 1952, Cordes et al 2016).

REVIEWERS' COMMENTS:

Reviewer #1 (Remarks to the Author):

Great job at addressing concerns.

Reviewer #2 (Remarks to the Author):

With their revision the authors addressed all points raised by me. In addition they updated the discussion and it is interesting to know that BCAT1 loss of function also impacts Irg1 in BCAA deprived medium suggesting a secondary activity of BCAT1 despite its transaminase activity.

Only two minor points left from this reviewer:

1. On page 7, line 158 of the revised manuscript and also in the initial submission, the authors show increased OCR induced by LPS. However, in the discussion (page 14, line 293) they state that respiration is decreased in M(LPS) macrophages contradicting their on measurements. Most likely, the discussion relates to Palsson-McDermott et al. or Tannahil et al. who observed decreased respiration in LPS activated macrophages. ERG240 reduces OCR.

Author's reply. The reviewer is correct. The papers cited (Palsson-McDermott et al. or Tannahil et al.) use murine BMDMs and our work show increased OCR in human macrophages. This point has now been clarified by deleting the statement on decreased respiration in the discussion (page 15 in the manuscript with track changes).

2. Other functions of itaconate: There are also studies out demonstrating that itaconic acid inhibits SDH (1960s) in general and more recently specifically in macrophages under inflammation (Booth et al 1952, Cordes et al 2016).

Author's reply. The reference suggested by the reviewer (Booth et al., 1952) is now added in the revised manuscript.